# Paper-Based Exosomal MicroRNA-21 Detection for Wound Monitoring: A Proof of Concept and Clinical Validation Trial Study

**DOI:** 10.3390/ijms24129822

**Published:** 2023-06-06

**Authors:** Shin-Chen Pan, Chi-Hung Lai, Van-Truc Vu, Cao-An Vu, Chun-Jen Huang, Chao-Min Cheng, Wen-Yih Chen

**Affiliations:** 1Department of Surgery, Section of Plastic and Reconstructive Surgery, National Cheng Kung University Hospital, College of Medicine, International Center for Wound Repair and Regeneration, National Cheng Kung University, Tainan 704, Taiwan; pansc@mail.ncku.edu.tw; 2Department of Chemical and Materials Engineering, National Central University, Taoyuan 32001, Taiwan; k2328993@gmail.com (C.-H.L.); vantruc1310@gmail.com (V.-T.V.); vucaoan@gmail.com (C.-A.V.); cjhuang@ncu.edu.tw (C.-J.H.); 3Institute of Biomedical Engineering, National Tsing Hua University, Hsinchu 300, Taiwan

**Keywords:** P-ELISA, exosome, microRNA-21, wound healing

## Abstract

Emerging evidence has shown that microRNAs play pivotal roles in wound healing. MicroRNA-21 (miR-21) was previously found to upregulate in order to fulfill an anti-inflammation role for wounds. Exosomal miRNAs have been identified and explored as essential markers for diagnostic medicine. However, the role of exosomal miR-21 in wounds has yet to be well studied. In order to facilitate the early management of poorly healing wounds, we developed an easy-to-use, rapid, paper-based microfluidic-exosomal miR-21 extraction device to determine wound prognosis in a timely manner. We isolated and then quantitatively examined exosomal miR-21 in wound fluids from normal tissues and acute and chronic wounds. Eight improving wounds displayed lower levels of exosomal miR-21 expression after wound debridement. However, four instances of increased exosomal miR-21 expression levels were notably associated with patients with poor healing wounds despite aggressive wound debridement, indicating a predictive role of tissue exosomal miR-21 for wound outcome. Paper-based nucleic acid extraction device provides a rapid and user-friendly approach for evaluating exosomal miR-21 in wound fluids as a means of monitoring wounds. Our data suggest that tissue exosomal miR-21 is a reliable marker for determining current wound status.

## 1. Introduction

Wound healing is a complex process that consists of multiple components, including cells and biomarkers, to participate in this event. Enhancement of wound healing has involved identifying and solving the underlying cause to allow the body to heal the wound naturally. Early detection of wound status via wound biomarker examination has the potential to prevent acute and chronic wounds from worsening. In addition to facilitating early clinical detection of symptomatic wounds, wound-related biomarkers serve as surrogate markers to predict wound outcomes and promote prompt therapeutic intervention. Some wound-fluid or tissue-derived factors, such as metalloproteinase enzymes (MMPs), growth factors, and interleukins, have been reported to serve as potential markers for predicting wound outcomes [1]. 

It is clear that microRNA (miRNA), involved in gene regulation to control cellular pathways, has become a powerful diagnostic tool with medical implications. The role of miRNAs in skin wound healing is well-established. MiRNAs implicated in keratinocyte proliferation, migration, and differentiation provided the opportunity for specific investigation of wound healing status [2]. MicroRNA-155 was reported to be highly expressed in the inflammatory phase of wound repair [3]. MicroRNA-21 (miR-21) is one of the early identified miRNAs that has been detected in various biological events through many of its gene targets [4]. MicroRNA-21(MiR-21) has also been shown to have an anti-inflammatory role and was upregulated in macrophages when alleviating inflammation [5]. Of note, the most studied miR-21 might have demonstrated its potential value as a diagnostic biomarker of wound healing status. Extracellular vesicles such as exosomes have been developed as important mediators for transmitting intercellular biological signals, including miRNAs, to modulate cellular function [6]. The role of exosomal miRNAs has been studied in cardiovascular diseases. It has been reported that miR-21 levels were significantly higher in acute myocardial infarction patients than those in healthy controls [7]. Further, exosomal miRNAs have been shown to act as better biomarkers than non-exosomal miRNAs for disease diagnosis [8]. However, using exosomal miR-21 to determine wound prognosis has not been studied until now. 

Paper-based ELISA (P-ELISA), a well-developed and convenient tool in diagnostic medicine, has been widely applied to examine the state and severity of various diseases [9,10]. We successfully identified high human neutrophil elastase expression levels in chronic wound fluids using P-ELISA [11]. Although nucleic acid testing (NAT) has been widely used for disease diagnosis, food safety control, and environmental monitoring, it currently requires laborious off-chip exosome separation and nucleic acid extraction prior to detection. Here, we developed an easy-to-use, rapid, paper-based microfluidic-extraction device to investigate wound status by analyzing exosomal miR-21 in wound fluids using reverse transcription-quantitative polymerase chain reaction (RT-qPCR). The results may be used to help determine wound prognosis and consequently provide better wound management.

## 2. Results

### 2.1. Development of Paper-Based Nucleic Acid Extraction Device

To detect a particular nucleic acid in our wound fluid samples, we developed a 3D paper-based nucleic acid extraction device by coating Whatman grade 1 filter paper with silica to reinforce nucleic acid binding (Figure 1). Nucleic acids were then adsorbed onto the silica nanoparticle surface using a binding solution with a pH value lower than the pKa value of the silica OH group (pH = 4) [12]. In this manner, Whatman grade 1 filter paper was made suitable for use as a paper-based immunoaffinity device and a paper-based nucleic acid extraction device. Hydrophobic and hydrophilic areas were clearly divided by All Purpose Duct Tape DT8 (3M™) to create different microfluidic device types. The specific exosomal anti-CD63 antibody was coated onto the device to isolate exosomes from wound fluids. Based on this design, sample fluid was added to the inlet port on the paper-based immunoaffinity device to capture exosomes by CD63 binding. After exosomal cleavage with lysis buffer, nucleic acids in the exosomes were adsorbed by the silica nanoparticle onto the paper-based nucleic acid extraction device in a salt buffer solution at pH higher than the pKa of silica surface OH groups or in dd water at an elevated temperature (55 °C). The morphology and size of captured exosomes were analyzed by scanning electron microscopy (SEM) and qNano, respectively. In SEM, the initial size of exosomes was about 100 nm with round shapes (unpublished data). The size distribution of exosomes was approximately 110–160 nm, characterized by the qNano [13]. However, the dimension of exosomes was expanded to the range of 200–600 nm after being treated with 95 °C ddH_2_O, with the maximal dimension up to more than 800 nm after the paper was added 10 µL of 95 °C ddH_2_O six times every 5 min (unpublished data). Thus, miR-21 could be released from exosomes in the wound fluid samples under this condition. We further used paper-based exosomal nucleic acid extraction devices and antibodies conjugated to horseradish peroxidase to produce a colorimetric readout. The miRNA expression levels could be quantified by RT-qPCR to assess wound status within 30 min.

In order to achieve rapid assessment of nucleic acid for point-of-care testing (POCT), a paper-based nucleic acid extraction device was developed to replace traditional nucleic acid extraction procedures. This approach requires no expensive or large instruments, and ultracentrifugation or ultrafiltration are not necessary. This paper-based approach easily handles exosome extraction and serves as an avenue for point-of-care testing development. In this process, Whatman grade 1 filter paper was cut to a suitable size. The filter paper was coated with 20 µL of Silanol solution (3 mg/mL in dH_2_O) on both sides and was desiccated at room temperature. In order to maintain clinical result consistency, all parameters such as adsorption time, elution time, temperature, and pH value used were fixed [12]. The prototype of the device was also shown in our previous study [13]. To improve the efficiency of the device and facilitate the maximal adsorption and elution ability of nucleic acid, the binding buffer was set at pH = 4. At room temperature, hydroxyl on the silanol surface reduced the negative charge repulsion force and provided more room for hydrogen binding with a nucleic acid. The adsorptive ability of nucleic acid was aided by a reduced electrostatic repulsive force and promoted by the hydrogen and possible salt bridge between nucleic acid and the silica-surface OH group, which was relatively stable in the pH 4 and room-temperature environment. To meet POCT criteria, we aimed to assess nucleic acid levels at room temperature. According to a previous study, the elution rate was better at 55 °C than at room temperature [14]. We then chose a pH 9 solution as our elution buffer and warmed it up to 55 °C for 45 min to increase deprotonation and negative charges on the Silanol surface. Thus, the elution of nucleic acid was made by the cleavage of hydrogen bonds between silica and nucleic acid.

We then chose miR-21 samples to study elution time. Elution time points for observation via electrophoresis were 15 min, 30 min, 45 min, and 60 min (Figure 2A). The data show that elution concentration increased with time and reached its highest capacity at 45 min (Figure 2B). To reduce the unstable pH value and examination bias of RT-qPCR caused by the tiny sample volume, we selected ddH_2_O as our binding and elution buffer. To determine optimal PCR reaction conditions, synthesized miR-21 was diluted into 109 copies/µL, 108 copies/µL, 107 copies/µL, 106 copies/µL, and 105 copies/µL and applied to the paper-based nucleic acid extraction device. Samples were bound with ddH_2_O for 3 min at room temperature and then desorbed for 45 min at 55 °C. A PCR calibration curve and an amplification plot were used to determine the quantitative levels of miR-21.

### 2.2. The Expression Levels of Exosomal miR-21 Were Higher Than Non-Exosomal miR-21 from Wound Fluids 

A previous study has shown that a majority of miRNAs are expressed in the exosomal serum of hepatocellular carcinoma patients [15] and exosomal saliva [16]. However, differences in miR-21 concentration in accordance with different wound conditions have not been previously addressed. We thus examined the differences between exosomal and non-exosomal miR-21 in wound fluids. Our data show that expression levels of exosomal miR-21 were higher than those of non-exosomal miR-21 from wound tissue (Figure 3). 

To further understand the role of exosomal miR-21 in clinical wound status, we used a paper-based nucleic acid extraction device to investigate exosomal miR-21 in wound fluids. We first tested three chronic wound samples originating from three different body parts (left leg, right leg, left elbow) of one patient with multiple simultaneous injuries. Samples were added directly to the antibody-coated zones without preprocessing, and RT-qPCR was once again used as a finished product evaluation tool. The obtained Ct values show negligible differences in exosomal miR-21 levels between the three samples (Ct values, left leg = 32.86 ± 0.49; Right leg = 32.85 ± 0.40; left elbow = 33.67 ± 0.29), indicating that exosomal miRNA-21 in wound fluid was independent of the injured location as long as the same physiological and treatment conditions were experienced.

### 2.3. Higher Exosomal miR-21 Is Associated with Poor Wound Healing

A previous study demonstrated that all significant miRNAs could be detected in circulating exosomes of tongue cancer patients, compared to less detectable miRNAs in patient plasma [17]. We, therefore, studied the value of exosomal miR-21 and non-exosomal, free miR-21 in wound tissue for wound assessment. Exosomal miR-21 was isolated from patients with normal tissue and acute and chronic wound fluids. Normal tissue displayed a wide range of exosomal miR-21 expression. There were no significant differences in miR-21 expression between normal tissue, acute, and chronic wounds (Ct values, normal tissue = 28.95 ± 1.29, 25.77–31.69, n = 11, acute = 29.73 ± 1.53, n = 36; chronic = 29.93 ± 1.17, n = 21, *p* = 0.71). This result is in keeping with a previous report indicating limitations in studying patient miRNAs due to such characteristics as age [18]. Although no significant differences in exosomal miR-21 expression were observed between acute and chronic wounds, we observed a differential exosomal miR-21 expression in the same patient in accordance with different wound statuses. This study also examined changes in wound tissue exosomal miR-21 among 13 patients before and after wound debridement. Eight improving wounds displayed lower levels of exosomal miR-21 expression after wound debridement. However, four cases of increased exosomal miR-21 expression levels were noticed in poor healing wounds despite aggressive wound debridement (Appendix A), indicating a potential role of exosomal miR-21 for wound outcome (same patient).

Mir-21 could accelerate epidermal growth factor (EGF) induced pancreatic cancer cell proliferation [19]. EGF plays a crucial role in skin wound healing. It could enhance wound healing through the stimulation of keratinocyte migration and acceleration of skin epithelialization [20]. Regulation of EGF by exosomal miR-21 in wounds is unknown. According to previous observations, we attempted to analyze the correlation between EGF and exosomal miR-21 in different wound statuses. We measured EGF expression levels in different wound fluids. A total of 92 clinical samples were analyzed using the traditional ELISA method. The data indicate that EGF expression levels in chronic wounds are significantly higher than those in acute wounds (acute = 49.06 ± 39.09 pg/mL, n = 49; chronic = 75.59 ± 42.35 pg/mL, n = 43, *p* < 0.05). Although the expression levels of EGF in chronic wounds were significantly higher than those in acute wounds, there was no strong correlation between EGF and exosomal miR-21 (R^2^ = 0.087) in wound fluid samples. Despite a weak relationship between EGF and exosomal miR-21 in our results, both exosomal miR-21 and EGF showed an assisted effect in evaluating clinical wound status. The expression of exosomal miR-21 and EGF were simultaneously investigated in eight patients with pre- and post-wound debridement. Three improving wounds had lower levels of exosomal miR-21 and EGF expressions compared to higher exosomal miR-21 and EGF values in poor healing wounds, suggesting a similar role of exosomal miR-21 and EGF in wound prognosis (Figure 4, Appendix A).

## 3. Discussion

With the increasing need for point-of-care wound assessment, we developed an inexpensive and easy-to-use paper-based nucleic acid detection device to investigate exosomal miR-21 in wound fluids that was superior to traditional methods that require expensive infrastructure and time-consuming processes. This paper-based nucleic acid extraction device provides a potent, user-friendly approach for capturing exosomes for analysis by SEM and qNano to examine morphology and size, respectively. When exosomal miR-21 was successfully extracted from wound fluids, a colorimetric readout was produced within 30 min. The data were analyzed using RT-qPCR to provide information for wound assessment. Our data suggest that exosomal miR-21 is a reliable marker for the early detection of wound status. The paper-based exosomal miR-21 extraction device developed in this study may be leveraged to produce a promising, first-of-its-kind screening tool for clinical wound monitoring.

MiR-21 is the most widely studied miRNAs in cancer and heart diseases [21,22]. MiR-21 could be found in the cytosol, exosome, neoplastic, and non-neoplastic cells [4]. As a result of interaction with cell membrane protein, exosomal miR-21 was less vulnerable to extracellular RNase degradation [21]. After activation, miR-21 could be released from the exosome and remain stable by natural leakage in the serum or body fluids [23]. In addition, a previous study demonstrated a similar expression profile for miR-21 between tissue and plasma samples from breast cancer patients [24], and most studies have shown that a majority of miRNAs are expressed in the exosomal serum of hepatocellular carcinoma patients [15] and exosomal saliva [16]. Sensitivity was higher when using exosomal miR-21 as a biomarker for diagnosing hepatocellular carcinoma compared to detection by serum miR-21 [25], which further supports our finding that exosomal miR-21, rather than non-exosomal miR-21 in wound tissue, might be a better prognostic marker for wound assessment. Furthermore, miR-21 was detected in different kinds of body fluids, such as urine, saliva, tears, and breast milk [26]. MiRNAs in body fluids might indicate a specific role associated with the surrounding tissues [26]. We showed here that exosomal miR-21 was exceptional stability present in the wound tissue fluids, indicating the development of exosomal miR-21 as a novel tool of tissue-based wound biomarkers.

MicroRNAs have recently been recognized as important regulators in skin wound healing [27]. They have been reported to regulate healing processes such as proliferation, migration, and angiogenesis [28]. MiR-21 was found to be upregulated in activated epithelial cells and mesenchymal cells and involved in wound contracture and collagen deposition in mice [29]. However, differences in miR-21 concentration in accordance with different wound conditions have not been previously addressed. Our study highlighted the value of exosomal miR-21in wound prognosis. A previous study demonstrated that higher miR-21 expression was related to the poor outcome of colorectal cancer patients [30]. Moreover, miR-21 was overexpressed in nonhealing venous ulcers [31]. Synthetic miR-21 was shown to inhibit epithelialization in a human skin organ culture wound model and to decrease granulation tissue formation in a rat model [31]. Consistent with that notion, our study suggests that higher exosomal miR-21 is associated with poor wound healing.

Various cytokines can regulate skin wound healing. Like EGF, miR-21 could also increase the proliferation and migration of keratinocytes to improve wound healing [4]. MiR-21 can activate MAPK/ERK signaling pathway to promote fibroblast activity and wound healing [4]. EGF-regulated MAPK/ERK signal cascade to induce cancer cell survival and proliferation [19]. Consistent with this notion, our observation of similar responses of mir-21 and EGF for wound status may suggest that both factors share similar molecular mechanisms in wound healing. Future studies should investigate how miR-21 regulates the EGF signaling pathway in the wound-healing process.

Our results demonstrate that paper-based devices are well suited for extracting tissue-derived exosomal miR-21 from wound fluids for the determination of wound status. Paper-based nucleic acid extraction device to isolate exosomes serves as an easy user interface in both the research and clinical applications, compared to the expensive and nonspecific process of the centrifugal method, which involves a series of centrifugations, filtration, and high-speed ultracentrifugation to obtain exosomes [32]. Furthermore, the magnetic beads technique can also be used to isolate exosomes. This method involved adherence to magnetic beads coated with antibodies and ensured obtaining high-quality exosomes. However, magnetic beads may affect the bioactivity and stability of exosomes [33]. Clinical validation of this device is, however, limited by the fact that low levels of wound exudate and complicating the wound environment may affect sample collection and the cross-section of exosomal miR-21 and substrate. Additionally, a temperature-controlled environment is a basic requirement for PCR amplification. Combining loop-mediated isothermal amplification with lateral flow assay or a field-effect transistor to examine exosomal miRNAs will allow our device to become more efficient and convenient and also increase the possibilities for clinical application. Additional investigations will be needed to confirm this hypothesis. Future studies should be conducted regarding the sensitivity and specificity of paper-based exosomal miR-21 extraction devices for detecting and monitoring various wound conditions and to meet the ASSURED (affordable, sensitive, specific, user-friendly, rapid and robust, equipment-free, and deliverable to end users) criteria of POCT device as suggested by WHO. In conclusion, we established a proof-of-principle paper-based extraction device for tissue-based exosomal miR-21 wound status detection by using clinical wound fluids from wound patients.

## 4. Materials and Methods

### 4.1. Patient Samples

In order to study miR-21 expression in wound fluids, a total of 68 tissue samples (normal: 11, acute: 36, chronic: 21) were harvested using standard surgical treatment procedures. The normal tissue samples were obtained from patients undergoing reconstructive surgery. All samples were mixed with protein extraction buffer (RIPA, Merck, Millipore, Temecula, CA, USA) and protein inhibitor (Merck, Millipore) and centrifuged at 4000 rpm for 30 min at 4 °C. Informed consent was obtained from all patients receiving the treatment, and all study procedures were approved by the Institutional Review Board at National Cheng Kung University Hospital (No. B-ER-110-506).

### 4.2. Paper Modification for Exosome Isolation

To isolate exosomes from multicomponent samples, we modified the surface of Whatman cellulose paper by adding anti-CD63 to specifically bind with tetraspanin protein CD63 on the exosomal membrane via specific and high-affinity antigen-antibody interaction. Briefly, circular pieces of Whatman cellulose chromatography paper (Grade 1) with a diameter of 6 mm were prepared as test zones and incubated in 50 μL of 3-mercaptopropyl trimethoxysilane (MPTMS) solution (4% in 99% ethanol) and N-γ-maleimidobutyryloxy succinimide ester (GMBS) solution (0.01 μmol/mL in 99% ethanol) for 30 min and 15 min, respectively. The papers were then incubated in 50 μL of NeutrAvidin solution (10 μg/mL in PBS) for 60 min at 4 °C. Afterward, 20 µL of 1% BSA (*w*/*v*) in PBS was added as a blocking solution 3 times every 10 min. Finally, 20 μL of biotinylated anti-CD63 antibody solution (20 μg/mL in PBS containing 1% (*w*/*v*) BSA) was dispensed onto the test zones 3 times every 10 min to bind with NeutrAvidin. Removing unbound molecules with three PBS washes provided us with anti-CD63-modified paper. Heterogeneous wound fluid samples (20 µL) were dispensed onto these coated papers, which were incubated for 20 min to capture exosomes by virtue of the specific antibody-antigen binding. Unwanted materials were washed off with PBS. All steps were performed at room temperature unless otherwise stated.

### 4.3. Paper-Based Procedure for Exosomal Nucleic Acid Extraction

Once isolated, exosomes were lysed by adding 95 °C ddH_2_O to the experimental area, whereupon the released nucleic acids were adsorbed with silica particles. Specifically, 10 μL of 95 °C ddH_2_O was slowly added to the exosome capture zones 6 times every 5 min, followed by the addition of 20 µL of silica solution (3 mg/mL in diH_2_O) to allow the silica particles to absorb the released nucleic acids. The experimental papers were then rinsed 3 times with 20 μL of ddH_2_O to remove unwanted materials before being immersed in a tube containing 20 µL of ddH_2_O (an equal volume to the originally used volume of sample) and incubated at 55 °C for 45 min for nucleic acid elution. After incubation and paper removal, the solution obtained represented the exosomal nucleic acid solution with the same concentration as the original sample.

### 4.4. Paper-Based Procedure for Free Nucleic Acid Collection

After adding chronic wound fluid to the antibody-coated paper to capture exosomes and using PBS to wash away unbound molecules, the wash solution was collected to analyze it for free nucleic acids. A piece of paper with dimensions of 0.5 mm × 3 mm was prepared, and silica particles were coated on the first third of the paper. The solution containing the obtained exosome-free nucleic acids was slowly added to the silica-coated paper. By leveraging the capillary phenomenon without the need for external force, the nucleic acids were absorbed onto the silica surface while other impurities followed the liquid flow to the rest of the paper. One-third of the paper containing nucleic acid-absorbing silica particles was cut off and dipped in 20 µL of ddH_2_O (an equal volume to the originally used sample volume) and incubated at 55 °C for 45 min for nucleic acid elution. After incubation and paper removal, the solution obtained represented the free nucleic acid solution with the same concentration as the original sample.

### 4.5. RT-qPCR for Nucleic Acid Quantification

RT-qPCR were used to analyze miRNA expression. These steps were performed using a Veriti™ Thermal Cycler and a StepOnePlus™ Real-Time PCR System (Applied Biosystems, Foster City, CA, USA), respectively. Experimental procedures and primer design were inspired, as previously reported [34]. In the RT reaction, each 10.2 µL reaction mixture contained 1.0 µL of ATP, 1.0 µL of dNTP, 1.0 µL of 10× Poly(A) buffer, 1.0 µL of reverse transcription primer, 0.5 µL of reverse transcriptase, 0.2 µL of *E. coli* Poly(A) polymerase, 0.2 µL of RNAse inhibitor, 4.3 µL of nuclease-free water, and 1.0 µL of RNA sample. The reverse transcription reaction temperature flow was 60 min at 42 °C, 5 min at 95 °C, and incubation was at 4 °C. The reverse transcription product containing synthesized cDNA was then amplified using the SYBR Green qPCR method. The 10 µL of SYBR Green qPCR mixture included 5.0 µL of SYBR Green I Master mix, 1.0 µL forward primer, 1.0 µL reverse primer, 2.0 µL nuclease-free water, and 1.0 µL of reverse transcription product. Thermocycling conditions were 5 min at 95 °C for initial heat activation and 40 cycles of 10 s at 95 °C and 30 s at 60 °C for annealing. Finally, Ct values were measured to represent the expression of the target miRNA.

Universal Reverse Transcription Primer: 5′-CAGGT CCAGT TTTTT TTTTV N-3.

miR-21 Forward Primer: 5′-TCAGT AGCTT ATCAG ACTGA TG-3′.

miR-21 Reverse Primer: 5′-CGTCC AGTTT TTTTT TTTTT TTCAA C-3′.

### 4.6. ELISA for EGF

Conventional plate ELISA (Elabscience, No.: E-EL-H0059) was used to measure the expression levels of EGF in wound fluids. Each specimen was applied in triplicate, and the average value was used as the final data. Values were expressed as mean ± SD.

### 4.7. Statistical Analysis

Analysis of variance was used to determine statistical differences of exosomal miR-21 among normal tissue, acute, and chronic wound fluids. Values of *p* < 0.05 were considered statistically significant. The results were expressed as mean ± SD.

## Figures and Tables

**Figure 1 ijms-24-09822-f001:**
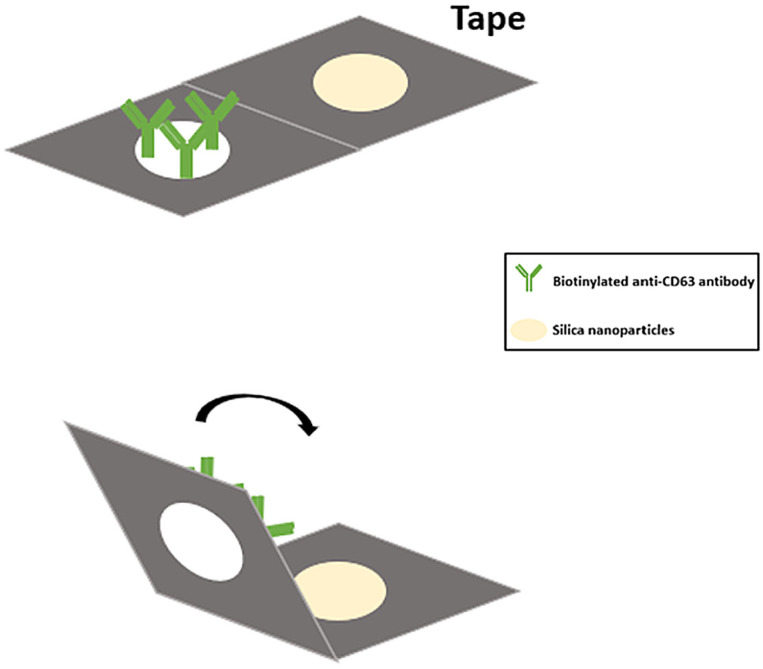
Three-dimensional paper-based nucleic acid extraction device to detect miR-21 in wound fluid.

**Figure 2 ijms-24-09822-f002:**
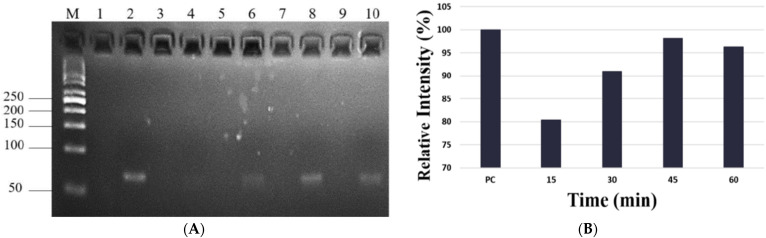
Comparison of different elution times of miR-21 extraction with electrophoresis: (**A**, left) Elution times for observation of miR-21 were 15 min, 30 min, 45 min, and 60 min. (**B**, right) Quantitative data of miR-21 expression levels at different elution times. The data showed that elution increased with time and reached its highest capacity at 45 min.

**Figure 3 ijms-24-09822-f003:**
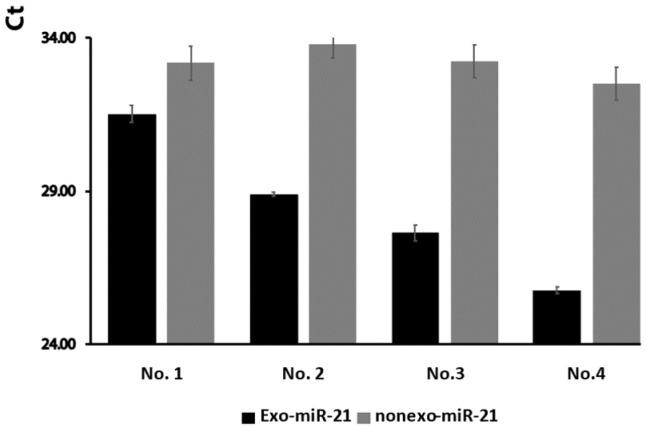
Comparison of expression levels of exosomal miR-21 and non-exosomal miR-21 in wound fluid. Exosomal miR-21 and non-exosomal miR-21 levels were investigated in 4 different wound samples. The data show that exosomal miR-21 expression level in wounds was higher than non-exosomal miR-21 expression level.

**Figure 4 ijms-24-09822-f004:**
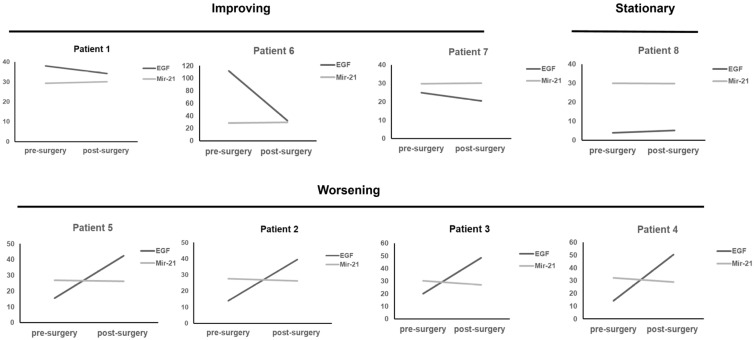
Expression level of miR-21 and EGF before and after wound debridement. Three improving wounds had lower levels of miR-21 and EGF expressions. Higher miR-21 and EGF values were observed in 4 poor healing wounds. The unit of miR-21 is Ct value and pg/mL for EGF.

## Data Availability

The data presented in this study are available on request from the corresponding authors.

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
