# Peer review of "Paper-Based Exosomal MicroRNA-21 Detection for Wound Monitoring: A Proof of Concept and Clinical Validation Trial Study"

_ijms, 2023, doi:10.3390/ijms24129822_

Round 1

Reviewer 1 Report

Paper-based Exosomal MicroRNA-21 Detection for Wound Monitoring: A Proof of Concept and Clinical Validation Trial  Study

Comments:

In this article, the authors have utilized a paper based microfluidic device for the extraction of exosomal miR-21 and subsequently evaluating for their relevance to wound monitoring.  

The authors could reconsider the following points, which will surely improve the quality of this manuscript.

1.      The word RT-PCR, and qPCR has been randomly used. I would suggest to consider proper mentioning of the terms and be consistent. The same would be considered for silicon oxide or silica.

2.      I suggest, that the authors should reconsidering writing the introduction with most relevant references from the latest literature.

3.      The claim of POCT device has to be justified with ASSURED criteria.

4.      Figure 2. Should be provided with better caption to follow the data.

5.      Throughout the assay procedure, PBS was used as wash buffer, why the authors didn’t consider using PBST?

6.      Any experiments or characterization performed to analyse the surface coverage of anti CD-63 antibodies and their stability over time?

7.      Page 4, Line 161, seems to be not scientific, statistical values or numerical is preferred over “some patient” with a better interpretations of the table.

8.      Page 2, Line 67, the reliance of pH has to be supported with characterization data. Page 3, lines 94-96 has to be provided with appropriate references.

9.      What is the size of the Silica particles used and their number density? How such properties influence the results?

Be consistent with the terms used in the manuscript.

Reviewer 2 Report

This study is interesting with clinical significance. Exosome is a popular candidate for cytotherapy and liquid biopsy. It is necessary to develop an easy-to-use method for detecting cargos in exosome. The authors put forward a new point of view for detecting exosomal microRNA-21. The followings are some comments to the authors.

Comments:

1.I suggest keeping the abbreviation of microRNAs consistent. For example, in line 42 “MiRNA-155 ” and in line 43 “MiR-21”.

2.If abbreviation had been defined in the text when used for the first time, abbreviation is recommended in the text below. For example, in line 58 “polymerase chain reaction (qPCR)” and in line 88 “ polymerase chain reaction (qPCR)” .

3.Please define all abbreviations in the text when used for the first time. For example, in line 77 SEM.

4.I suggest that isolated exosomes by paper-based nucleic acid extraction device should be identified by electron microscopy, biomarker (such as CD9, CD63, CD81 and so on), and qNano.

5. I suggest that the exosomes isolated by paper-based nucleic acid extraction device  should be compared with exosomes isolated by traditional methods (such as centrifugal method).

6. Futher researches to study the relationship of exosomal microRNA-21 and poor healing wounds is needed. I suggest studying the relationship above at the cellular level and the animal level.

7.miR-21 in Table 1 is total miR-21 or exosomal microRNA-21? Please confirm that.

8. In figure 3, the sample size of 4 is small, the evidence is weak, so it is suggested increasing the sample size.

Round 2

Reviewer 1 Report

The authors have modified the manuscript according the comments provided. However, still the certain points are concerning to further improvise the quality manuscript,

1. The ASSURED criteria of POCT device as suggested by WHO is still needs to be addressed, in specific sensitivity and the specificity. A statement about the sensitivity and specificity needs to be included. 

2. Data regarding the use of PBST or appropriate citation needs to be provided in the supporting information. 

3. The storage data needs to be included in the manuscript.

4. The response to the size and number density of the particle is not satisfying. I assume the size of the nanoparticle is going to influence the surface coverage of nucleic acids, and the total number of particles involved determines the distribution of nucleic acid on their surface. Moreover, the claim of pH is not supported will. The pH claim can be supported by FTIR analysis using silica chips, while the number density and the adsorption ration could be determined through DLS or UV-vis spectroscopy. These data will significantly improve the quality of the manuscript. 

5. The photographic image of the paper strip showing different regions could be interesting to see.

Minor grammatical errors need to be corrected.

Reviewer 2 Report

I suggest this manuscript can be accepted in present form.

Author Response

Thanks for your comments.

Round 3

Reviewer 1 Report

No comments.